# Neoadjuvant Chemotherapy plus Interval Cytoreductive Surgery with or without Hyperthermic Intraperitoneal Chemotherapy (NIHIPEC) in the Treatment of Advanced Ovarian Cancer: A Multicentric Propensity Score Study

**DOI:** 10.3390/cancers15174271

**Published:** 2023-08-26

**Authors:** Antoni Llueca, Maria Victoria Ibañez, Pedro Cascales, Antonio Gil-Moreno, Vicente Bebia, Jordi Ponce, Sergi Fernandez, Alvaro Arjona-Sanchez, Juan Carlos Muruzabal, Nadia Veiga, Berta Diaz-Feijoo, Cristina Celada, Juan Gilabert-Estelles, Cristina Aghababyan, Javier Lacueva, Alicia Calero, Juan Jose Segura, Karina Maiocchi, Sara Llorca, Alvaro Villarin, Maria Teresa Climent, Katty Delgado, Anna Serra, Luis Gomez-Quiles, Maria Llueca

**Affiliations:** 1Multidisciplinary Unit of Abdominopelvic Oncology Surgery (MUAPOS), University General Hospital of Castellon, 12004 Castellon, Spain; ana_maiocchi@gva.es (K.M.); llorca_sarcar@gva.es (S.L.); villarin_alv@gva.es (A.V.); maclimen@uji.es (M.T.C.); delgado_kat@gva.es (K.D.); serraa@uji.es (A.S.); quiles@uji.es (L.G.-Q.); 2Oncological Surgery Research Group (OSRG), Department of Medicine, University Jaume I (UJI), 12071 Castellon, Spain; 3Department of Mathematics, IMAC, University Jaume I (UJI), 12071 Castellon, Spain; mibanez@uji.es; 4Department of General Surgery, Hospital Universitario Virgen de la Arrixaca, El Palmar, 30120 Murcia, Spain; pedroantonio.cascales@um.es; 5Gynecologic Oncology Unit, Department of Gynecology, Hospital Universitari Vall d’Hebron, Universitat Autònoma de Barcelona, 08035 Barcelona, Spain; antonio.gil@vallhebron.cat (A.G.-M.); vicente.bebia@vallhebron.cat (V.B.); 6Department of Gynecology, University Hospital of Bellvitge, 08907 Barcelona, Spain; jponce@bellvitgehospital.cat (J.P.); sfernandezg@bellvitgehospital.cat (S.F.); 7Unit of Surgical Oncology and Pancreatic Surgery, University Hospital Reina Sofia, 14004 Cordoba, Spain; alvaro.arjona.sspa@juntadeandalucia.es; 8Department of Gynecologic Oncology, Complejo Hospitalario de Navarra, 31008 Pamplona, Spain; jc.muruzabal.torquemada@navarra.es (J.C.M.); nadia.veiga.canuto@navarra.es (N.V.); 9Gynecologic Oncology Unit, Clinic Institute of Gynecology, Obstetrics, and Neonatology, Hospital Clinic of Barcelona, Institut d’Investigacions Biomèdiques August Pi i Sunyer (IDIBAPS), Universitat de Barcelona, 08036 Barcelona, Spain; bdiazfe@clinic.cat (B.D.-F.); celada@clinic.cat (C.C.); 10Department of Obstetrics and Gynecology, University General Hospital of Valencia, 46014 Valencia, Spain; gilabert_juaest@gva.es (J.G.-E.); aghababyan_cri@gva.es (C.A.); 11Unit of Peritoneal Carcinomatosis, Department of General Surgery, University General Hospital of Elche, 03203 Elche, Spain; fj.lacueva@umh.es (J.L.); a.calero@umh.es (A.C.); 12Hepatobiliopancreatic Surgery and Peritoneal Oncology Surgery Unit, General Surgery and Digestive System Department, Son Espases University Hospital, 07120 Palma de Mallorca, Spain; juan.segura@ssib.es; 13Department of General Surgery, University General Hospital of Castellon, 12004 Castellon, Spain; 14Department of Obstetrics and Gynecology, Joan XXIII University Hospital of Tarragona, 43005 Tarragona, Spain; mllueca.hj23.ics@gencat.cat

**Keywords:** hyperthermic intraperitoneal chemotherapy, advanced ovarian cancer, neoadjuvant chemotherapy, complete surgery, prgresion free survival, overall survival

## Abstract

**Simple Summary:**

Advanced ovarian cancer (Stages III-IV) continues to be one of the gynecological tumors with the highest mortality. Standard treatment consists of debulking surgery and subsequent adjuvant chemotherapy. Recently, some authors have postulated that the administration of hyperthermic chemotherapy during surgery could increase the survival of patients, especially in cases in which chemotherapy had already been administered before surgery to reduce tumor volume. Our study is important because it collects data from 11 tertiary hospitals in Spain, and the data are subjected to a statistical technique that reproduces the data that we would find in a prospective study but using retrospective data (propensity score matching). It also offers a current view of the status of ovarian cancer treatment in our country.

**Abstract:**

Introduction: Epithelial ovarian cancer (EOC) is primarily confined to the peritoneal cavity. When primary complete surgery is not possible, neoadjuvant chemotherapy (NACT) is provided; however, the peritoneum-plasma barrier hinders the drug effect. The intraperitoneal administration of chemotherapy could eliminate residual microscopic peritoneal tumor cells and increase this effect by hyperthermia. Intraperitoneal hyperthermic chemotherapy (HIPEC) after interval cytoreductive surgery could improve outcomes in terms of disease-free survival (DFS) and overall survival (OS). Materials and Methods: A multicenter, retrospective observational study of advanced EOC patients who underwent interval cytoreductive surgery alone (CRSnoH) or interval cytoreductive surgery plus HIPEC (CRSH) was carried out in Spain between 07/2012 and 12/2021. A total of 515 patients were selected. Progression-free survival (PFS) and OS analyses were performed. The series of patients who underwent CRSH or CRSnoH was balanced regarding the risk factors using a statistical analysis technique called propensity score matching. Results: A total of 170 patients were included in each subgroup. The complete surgery rate was similar in both groups (79.4% vs. 84.7%). The median PFS times were 16 and 13 months in the CRSH and CRSnoH groups, respectively (Hazard ratio (HR) 0.74; 95% CI, 0.58–0.94; *p* = 0.031). The median OS times were 56 and 50 months in the CRSH and CRSnoH groups, respectively (HR, 0.88; 95% CI, 0.64–1.20; *p* = 0.44). There was no increase in complications in the CRSH group. Conclusion: The addition of HIPEC after interval cytoreductive surgery is safe and increases DFS in advanced EOC patients.

## 1. Introduction

Epithelial ovarian cancer (EOC) is the leading cause of death in women with gynecological tumors, and in many cases, the disease is in an advanced stage at the time of diagnosis (International Federation of Gynecology and Obstetrics [FIGO] stage IIIC–IV) [1].

Cytoreductive surgery (CRS) and routine chemotherapy have been the main treatments for patients with ovarian cancer since the mid-1990s, with the removal of all macroscopically visible tumors being the strongest prognostic factor [2].

In ovarian cancer patients, the tumors are primarily confined to the peritoneal cavity, involving the peritoneum and the dissemination and implantation of tumor cells from the ovaries. The peritoneum-plasma barrier makes it difficult for chemotherapeutic agents to reach the peritoneum [3,4,5,6].

Therefore, the intraperitoneal administration of chemotherapy could better target disseminated peritoneal tumor cells and could improve outcomes by eliminating residual microscopic peritoneal tumor cells. In addition, compared to the intravenous administration of chemotherapeutic agents, difficulty in the absorption of chemotherapeutic agents through the peritoneum could reduce plasma toxicity [7] and increase the drug effect when delivered with hyperthermia (hyperthermic intraperitoneal chemotherapy [HIPEC]) [8].

Therefore, interest in HIPEC for ovarian cancer treatment in recent years has increased considerably, with many published studies showing promising results [7,9,10,11,12,13].

In 2018, Van Driel et al. [14] published the first randomized controlled trial (RCT) investigating the effect of HIPEC in primary EOC patients: 245 patients who were previously treated with three cycles of neoadjuvant chemotherapy were randomized to receive interval CRS or interval cytoreductive surgery plus HIPEC (CRSH); disease-free survival (DFS) and overall survival (OS) were significantly higher in patients who received HIPEC (10.7 vs. 14.2 months, *p* = 0.003 and 33.9 vs. 45.7 months *p* = 0.02, respectively).

These results have recently been confirmed in two recent RCTs. Cascales et al. [15] randomized 71 patients to receive interval CRS (control arm) or interval CRSH (experimental arm). The median OS times were 45 and 52 months in the control and experimental groups, respectively. The findings showed HIPEC to be an independent protective factor against the development of recurrence (hazard ratio [HR], 0.12, 95% confidence interval [CI], 0.02–0.89; *p* = 0.038). In the RCT by Lim et al. [16], HIPEC was shared in two subgroups, as first-line during first-intention CRS and in interval surgery after neoadjuvant therapy. In the first case, there were no improvements in the survival of the patients, but in the interval CRS group, improvements were observed both in progression-free survival (PFS) (17.4 vs. 15.4 months) ((HR for disease progression or death, 0.60; 95% CI, 0.37–0.99; *p* = 0.04)as in OS (61.8 vs. 48.2 months) (HR, 0.53; 95% CI, 0.29–0.96; *p* = 0.04)for the HIPEC group respect to the control group.

The main objective of this observational multicenter study was to investigate whether HIPEC administration after interval CRS improves outcomes in terms of DFS and OS compared to interval CRS alone in patients with advanced stages of EOC.

## 2. Materials and Methods

### 2.1. Study Design

A multicenter, retrospective observational study of advanced EOC patients who underwent interval cytoreductive surgery alone (CRSnoH) or interval CRSH was carried out in 10 tertiary hospitals in Spain between July 2012 and December 2021. All hospitals were referral hospitals for the treatment of ovarian cancer. A total of 515 patients were included in the study, but only 489 were ultimately selected due to the lack of information in the hospital records.

### 2.2. Patient Inclusion and Exclusion Criteria

In this study, we aimed to evaluate patients with a diagnosis of primary EOC, tubal carcinoma, or primary peritoneal carcinoma (International Federation of Gynecology and Obstetrics [FIGO] stage III/IV) who had been treated with three cycles of systemic neoadjuvant chemotherapy (NACT) because their abdominal disease was too extensive for primary CRS or they were not fit for primary surgery.

All patients had adequate bone marrow function (e.g., absolute neutrophil count ≥ 1000/mm^3^, platelet count ≥ 100,000/mm^3^, hemoglobin level ≥ 8.5 g/dL), renal function (e.g., creatinine level ≤ 1.5), normal hepatic function and normal blood coagulation parameters (e.g., prothrombin time with an international normalized ratio of ≤1.5). Patients must have also attained an Eastern Cooperative Oncology Group performance status of 0–2.

Subjects with low-grade or noninvasive disease, a nonovarian malignancy, or evidence of another cancer within the past 3 years were excluded from the study.

This retrospective study received institutional review board approval (CEIm number 2862020).

In the preoperative study, the tumor burden that the patients presented was quantified by assessing the Peritoneal Cancer Index (PCI) [17].

The PCI score was determined for all patients in the current study by preoperative thoraco-abdominal computed tomography and/or laparoscopy. To quantify the radiological PCI score, the largest tumor implanted in the assessed region was chosen and assigned a score of 0–3. The sum of the scores for each region was then used to calculate the radiological PCI score. All patients with a PCI score > 20 and those with abdominal disease determined to be too extensive for primary CRS at the discretion of the different surgical teams received NACT [18].

The PCI score was also calculated before and during surgery and was categorized into three ordinal levels: 1–10, 11–20, and >20.

Complete cytoreductive surgery (CCS) was defined as surgery that resulted in no visible disease (residual disease classification, R-1), optimal cytoreductive surgery (OCS) was defined as surgery that resulted in the presence of one or more residual tumors measuring less than 10 mm in diameter, and suboptimal cytoreductive surgery (SCS) was defined as surgery that resulted in the presence of one or more residual lesions measuring more than 10 mm in diameter.

Patients treated with HIPEC (75–100 mg/m^2^ cisplatin or 60 mg/m^2^ placlitaxel) underwent perfusion using an open technique with a target temperature of 41.5 °C for 60–90 min. The administration of HIPEC was performed using an open technique in the majority of cases, and in a small percentage, the technique was closed. Postoperative complications were described according to the Clavien–Dindo classification. Grade III–IV complications were considered major complications [19].

### 2.3. Disease-Free Survival and Overall Survival

DFS was defined as the length of time from the date of surgery until clinical, radiological, or CA-125 progression. OS was defined as the time from the date of surgery until death, with all causes of death treated equally. If a subject had not progressed or died, PFS and OS were censored at the time of the last follow-up.

### 2.4. Statistical Analysis

Given the nonrandomized nature of the study, analysis was first performed on all the data sets, describing and comparing the distribution of the different variables among the patients treated with and without HIPEC. The variables were summarized according to their nature with means and standard deviations (SDs) or with frequencies and percentages. To compare their distributions, parametric (*t* test) or nonparametric tests (Mann–Whitney test) were applied depending on whether the analyzed variable did or did not follow a Gaussian distribution. Chi-squared or Fisher’s exact tests were used for the qualitative variables.

Analyses of DFS and OS times were performed to detect factors that affect them. Kaplan–Meier estimates were compared among the different categories of each factor analyzed using the log-rank test, and HRs and their corresponding 95% CIs were estimated and compared to detect risk factors.

To avoid unnecessary artifacts, the series of patients who underwent CRSH or CRSnoH must be balanced regarding the set of risk factors in the study. Cox proportional hazard models are used to control for possible confounding effects that could weigh down the results. The complete way to carry out the above adjustment is based on the use of the propensity score (PS) or propensity indices in the statistical analysis of the data, which emulates the effects of a posteriori randomization of the confounding variables (risk factors) previously recognized, defined, and collected in the data table (quasi-randomization). Optimal 1:1 PS matching [20] analysis was performed.

The distribution of the variables was again checked for the matched subsample, and recurrence-free and overall survival times were again analyzed for this subsample to detect factors for which the survival differed between the two treatments (CRSH and CRSnoH).

Statistical analysis was performed using R (R Core Team, 2021. Vienna, Austria. Available at: https://www.R-project.org/) (accessed on 8 July 2022) [21]. The “MatchIt” [22], “cobalt” [23], “survival” [24], “EquiSurv” [25], and “survminer” [26] packages were used to analyze the PS matching and the survival function. Unless otherwise stated, all analyses were performed with a two-sided significance level of 0.05.

## 3. Results

From July 2012 to December 2021, a total of 515 patients at 10 participating centers in Spain were included. Twenty patients were excluded due to a lack of information.

### 3.1. Unmatched Series

The final unbalanced patient cohort included 489 advanced ovarian cancer patients. The mean age (percentile 25–75) at diagnosis was 61 (54–69) and 60 (53–68) years in the CRSH and CRSnoH groups, respectively. A total of 451 tumors were of high-grade serous histology, with 259 (91.2%) and 192 (97%) in CRSnoH and CRSH cohorts, respectively. The FIGO stages were predominantly stage III (153 (53.3%) in the CRSnoH group and 164 (81.2%) in the CRSH group).

The median PCI value of the whole unmatched group was 9, with median values of 9 and 10 in the CRSnoH and CRSH groups, respectively. The complete surgery rate was similar in both groups: 225 (78.9%) in the CRSnoH and 171 (84.7%) in the CRSH group. There were no differences in the major complications between the two groups of unmatched patients.

In patients treated with HIPEC, 95 (47%) received cisplatin, and 107 (53.0%) received paclitaxel postoperatively.

The global mean follow-up time was 39 months, with an SD of 27 months, and was 34.6 and 45.3 months (SD 22 and 32 months) for the CRSnoH and CRSH groups, respectively. Significant differences were observed between the groups, with the mean follow-up time being significantly higher in the CRSH group (*p* value < 0.001).

The demographic and baseline disease characteristics of the unmatched cohort are shown in Table 1.

When performing the survival analysis on the complete database, the factors that were observed to affect survival (DFS and/or OS) were PCI score, histological grade, histological subtype, and FIGO stage (Figure 1), so we balanced the two groups with respect to these variables.

Figure 1a,b show that different PCI Ievels are associated with an increased risk of recurrence and death. Therefore, patients with PCI values between 11 and 20 or over 20 have an increased risk of recurrence and death compared to those with PCI values under 10 (reference). Compared to the reference group, patients with PCI values between 11 and 20 had a recurrence risk that was increased by a factor of 1.47 (47%) and a death risk that was increased by a factor of 1.66 (66%); patients with PCI values greater than 20 had a recurrence risk that was increased by a factor of 1.91 (91%) and a death risk that was increased by a factor of 2.33 (133%). Compared with patients with histological grades G1–G2 (reference), those with grade G3 had a recurrence risk that was increased by a factor of 1.46 (46%). In the analysis of OS, the CI for the HR in patients with a histological grade G3 ranged from 0.9 to 2.33, which did not show a significant difference from the reference group. Regarding histological type, patients with clear cell carcinoma presented a greater risk of recurrence (HR 3.41) and death (HR 8.45) than those with serous carcinoma (reference). Patients with endometroid carcinoma also presented a greater risk of death (HR 3.73) than those with serous carcinoma. Finally, patients with FIGO stage IVA also presented a greater risk of death (HR 1.88) than patients with FIGO stage III, and patients with optimal surgery presented a greater risk of death than patients with complete surgery (HR 1.43).

### 3.2. Matched Series

After balancing the series with respect to the described variables (Figure 1), 170 patients were obtained from each subgroup (CRSH and CRSnoH), and the baseline demographic characteristics were similar between the two groups. The Love plot in Figure 2 summarizes the covariate balance, showing the mean difference of the distribution of each variable in both treatments on the initial data set (all; white dots in Figure 2) and the matched subsample (matched; black points in Figure 2). The dotted lines represent the threshold such that if most or all of the points after matching were within the threshold, good evidence suggests that balance has been achieved. In this case, a balance was reached for all the covariates except the histological grade. No other important intergroup differences in operative procedures or surgical outcomes were found.

For the clearest overview of the matched subsample, the clinicopathological characteristics of the included patients are described in Table 2.

### 3.3. Survival Analysis

In the matched series, the median DFS time was 16 months (IQR 9–40 months) in the CRSH group and 13 months (IQR, 8–26 months) in the CRSnoH group (HR 0.74; 95% CI, 0.58–0.94; *p* = 0.031) (Figure 3) (Table 3).

The median OS in the balanced series was 56 months (IQR, 36-NA months) in the CRSH group and 50 months (IQR, 29–87 months) in the CRSnoH group (HR, 0.88; 95% CI, 0.64–1.20; *p* = 0.44) (Figure 4) (Table 4).

In the assessment of patients with post-neoadjuvant HIPEC, the results of the subgroup analyses for DFS (Figure 5) showed a benefit for older age (HR 0.67; 95% CI: 0.45–1.00; *p* = 0.05), stage III FIGO classification (HR, 0.76; 95% CI: 0.58–1.00; *p* = 0.05), a high-grade serous histological type (HR, 0.75; 95% CI: 0.58–0.96; *p* = 0.023), comorbidities (HR, 0.66; 95% CI: 0.45–0.97; *p* = 0.04), and complete surgery (HR, 0.66; 95% CI: 0.48–0.97; *p* = 0.03).

In the evaluation of patients with post-neoadjuvant HIPEC, baseline characteristics, including age, stage, histological type, type of surgery, and the PCI score, were not significant for survival outcomes in the multivariable Cox proportional hazards regression model (Figure 6).

## 4. Discussion

Based on the results of this multicentric retrospective study, after adjusting for potential confounding variables with a PS matching analysis, the disease-free interval between surgery and the first relapse was improved in the group of patients who received NACT and interval debulking surgery plus HIPEC (Figure 1). A similar tendency was observed with OS (time from diagnosis to death) (Figure 2), but this was not statistically proven.

The findings of our study are consistent with the increased disease-free interval reported in the Van Driel trial [14] as well as in the Cascales and LIM studies [15,16], despite the heterogeneity of the drugs and doses administered in our study.

Hyperthermic intraperitoneal chemotherapy is considered a local or regional treatment for intraperitoneal disease. We have long known that residual tumors in the abdominal cavity are undoubtedly the most important prognostic factor for PFS and OS in patients with advanced ovarian cancer [27].

The control of intraperitoneal abdominal disease is very important. In 2008, Armstrong [28] showed that CRS combined with intraperitoneal chemotherapy was associated with increased survival. Therefore, local intraperitoneal control using surgery and chemotherapy with or without hyperthermia could improve survival outcomes.

Hyperthermia per se also induces alterations in the tumor microenvironment, producing alterations in vascularization and the oxygen supply to tumor cells. Furthermore, hyperthermia targets multiple DNA repair pathways, which are generally upregulated in cancer cells and protect them from DNA-damaging agents [29].

With this background and knowing that the definition of complete surgery in advanced ovarian cancer assumes the existence of microscopic disease, it seems reasonable to include the combination of maximal effort CRS and hyperthermic intraperitoneal chemotherapy in the control of intraperitoneal disease.

In view of the results of Lim [16], it seems that the application of HIPEC in patients with ovarian cancer who have not received chemotherapy does not seem as effective. Perhaps the results of the OVHIPEC II trial [30] will provide us with more data in this regard.

Moreover, one question remains unresolved in reviewing the results: Why is HIPEC effective only after recent exposure to chemotherapy and not in chemotherapy-naive women with ovarian cancer? The RCT of Spiliotis et al. [31] and some other retrospective studies [32] demonstrated the effectiveness of HIPEC in patients with ovarian cancer recurrence after treatment with chemotherapy.

The answer to this question remains unknown, but there could be various explanations for why HIPEC only works after recent chemotherapy.

Recently, ovarian stem cells have been identified by observing the renewal of postnatal follicles on the surface of the ovaries, which suggests the existence of so-called ovarian stem cells [33].

As previously stated, cytoreduction surgery and subsequent chemotherapy achieve a high percentage of clinical remission [34]. However, many patients will present with tumor recurrence. Such recurrence and chemoresistance of ovarian cancer could be explained by a cancer stem cell model [35].

Clinically, the possibility of survival and the recurrence of ovarian tumor cells after chemotherapy is possibly due to the persistence of ovarian cancer stem cells. This suggests that chemotherapy selects for highly aggressive ovarian cancer stem cells [36,37].

These chemoresistant ovarian tumor stem cells may be hidden in the normal-appearing or scarred peritoneum that appears after neoadjuvant therapy for ovarian cancer [38,39].

Some previous studies have already demonstrated the difficulty with which chemotherapeutic agents reach the peritoneum, where despite doubling the administered dose and with important side effects, no benefit was observed for the patients [40].

Thus, in the context of treatment after neoadjuvant chemotherapy, HIPEC could be a more than reasonable option to more effectively eliminate the subpopulation of resistant cancer stem cells embedded in the peritoneum [41].

Regarding hyperthermia, in a Cochrane review, it was recently postulated that inhibition of poly(ADP-ribose) polymerase (PARP)-1 after chemotherapy prolongs DFS in patients with EOC; however, the benefit has not been demonstrated or is very slight in terms of OS [42].

Currently, we know that defective homologous recombination DNA repair imposed by BRCA1 or BRCA2 deficiency sensitizes cells to and is currently used in the treatment of ovarian cancer. Some authors have reported that mild hyperthermia (41–42.5 °C) induces BRCA2 degradation and inhibits homologous recombination. Thus, hyperthermia per se can be used to sensitize innately HR-competent tumor cells to PARP-1 inhibitors [29,43].

On the other hand, the fact that the PFS was better in patients treated with HIPEC, especially in patients with complete surgery, reinforces the idea that the attempt to achieve a microscopic residual tumor, is the cornerstone of any treatment established for advanced EOC.

As already stated, aggressive surgery to eliminate microscopic residual tumors carries an increased risk of early postoperative complications and possibly compromises the patient’s prognosis [44,45,46,47]. Moreover, the addition of intraperitoneal chemotherapy has also been reported as a factor for increased postoperative complications [48]. In our study, there were no differences in the appearance of early complications or in 30-day postoperative mortality.

### 4.1. Strengths and Weakness

There are some weaknesses in this study, including its retrospective nature. It is well known that a retrospective study has multiple biases that limit its clinical validity, but it is also true that well-performed propensity score matching, by equally balancing the arms of the study, makes the results obtained using this technique resemble those of a prospective study. Another limitation of this study is the multiple care protocols of the different hospitals in this study, which can affect the results (nutritional status, days of admission, etc.). This is of special relevance when evaluating different HIPEC regimens (drugs, time, temperature) in the cohort of CRSH patients.

One of the main strengths of this study is that it offers a vision of the status of advanced ovarian cancer management in a recent period in Spain. Another strength of this study is that, for the first time, a representative group of patients with advanced ovarian cancer treatment from different surgical specialties (Spain Gynecology Oncology Group (Spain GOG) and the Spanish group of peritoneal oncology surgery (GECOP group)) has been included to evaluate the surgical treatment of advanced ovarian cancer in recent years.

Of the more than five hundred women with advanced ovarian cancer in this study, almost 40% were treated by general surgeons who specialized in peritoneal oncology surgery (most of these patients were included in the CRSH group). One of the possible reasons for this situation may be the nonexistence of the oncological gynecology subspecialty in Spain. We hope that this situation can be reversed with the accreditation of new specialists promoted from the ESGO (European Society of Gynecology Oncology) fellowship in gynecology oncology in which some of the tertiary hospitals included in this study have already been involved.

Another strength of this study is the considerable amount of statistical work carried out to balance and extract reliable conclusions from the data.

### 4.2. Future Perspectives

As future investigations of the application of HIPEC in advanced ovarian cancer, it would be interesting to assess the effect of this treatment in patients undergoing complete surgery compared to those undergoing suboptimal surgery and exclusively in patients with high-grade serous histological subtypes, which seems to be for whom HIPEC produces the greatest effect.

## 5. Conclusions

The addition of HIPEC after interval CRS following NACT increases the time from surgery to recurrence in patients with stage III or IV EOC. Hyperthermic intraperitoneal chemotherapy could be performed safely after maximal CRS without increasing the morbidity or mortality associated with the procedure. Treatment with HIPEC after optimal CRS should be seriously considered for these patients.

## Figures and Tables

**Figure 1 cancers-15-04271-f001:**
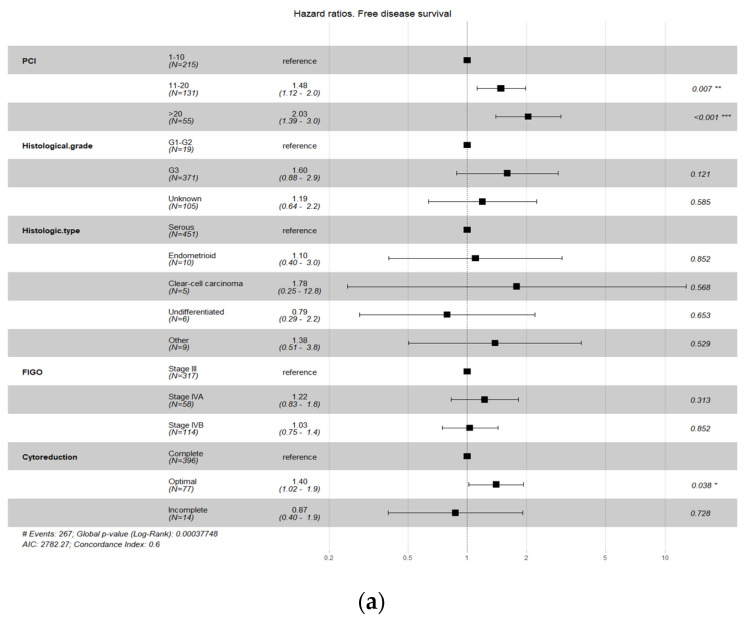
Principal factors affecting survival (DFS (**a**) and/or OS (**b**)). *, **, *** means different grade of statistical significance.

**Figure 2 cancers-15-04271-f002:**
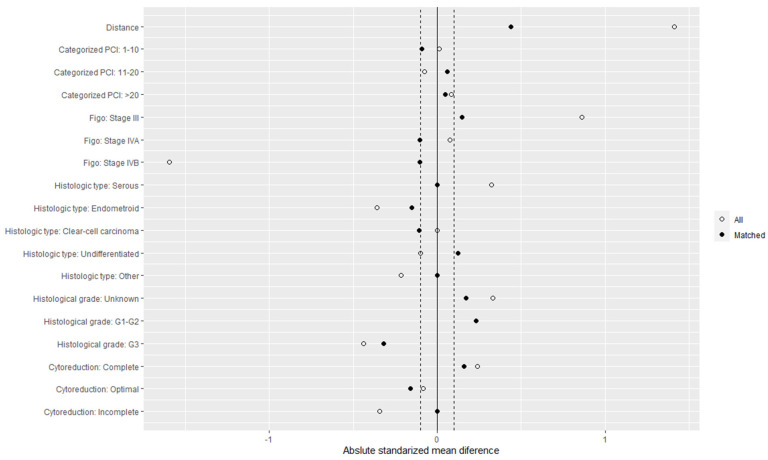
Love plot to visualize the goodness of the match.

**Figure 3 cancers-15-04271-f003:**
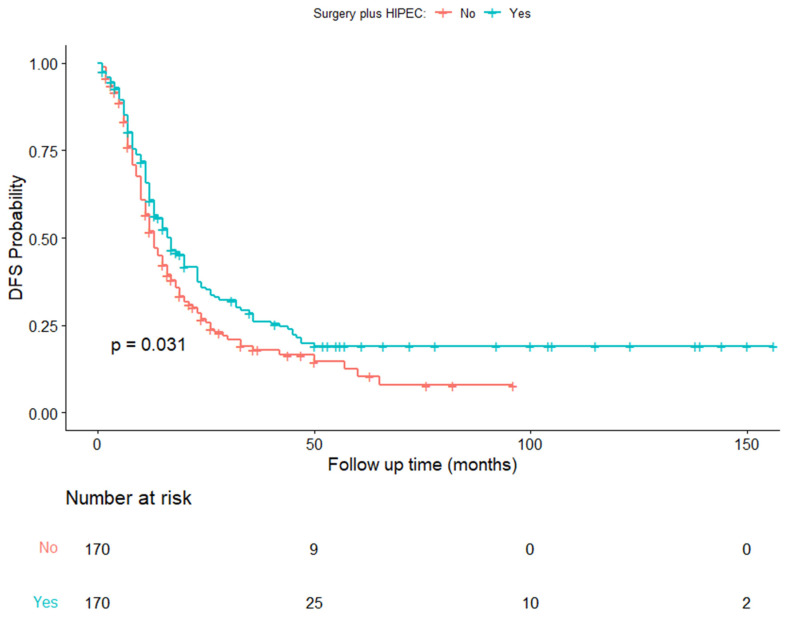
Kaplan–Meier estimates of disease-free survival in the time from surgery to relapse.

**Figure 4 cancers-15-04271-f004:**
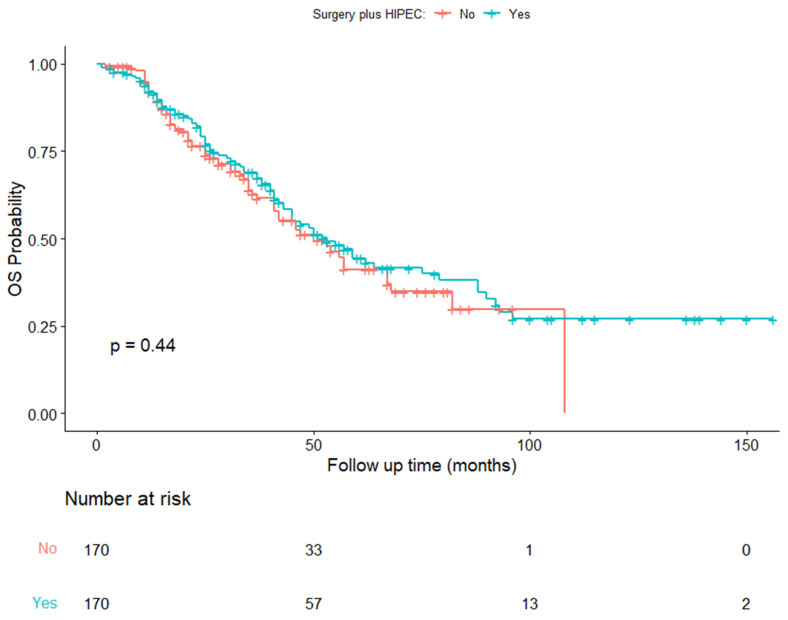
Kaplan–Meier estimates of overall survival in the time from diagnosis to death.

**Figure 5 cancers-15-04271-f005:**
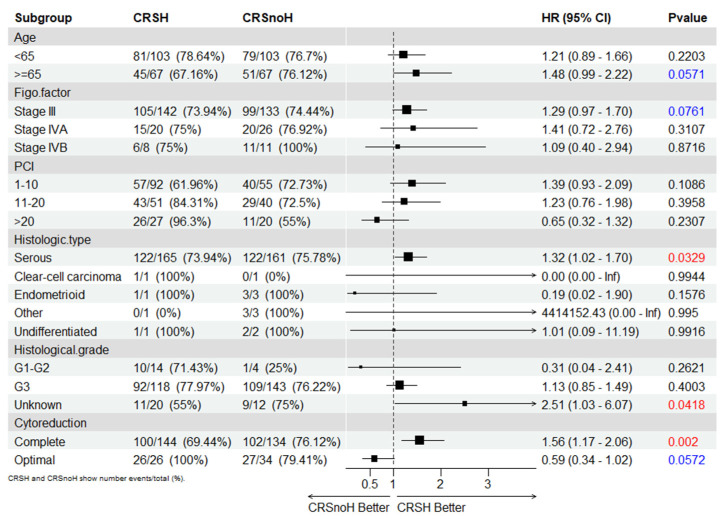
Subgroup analyses for the progression-free survival of the matched series.

**Figure 6 cancers-15-04271-f006:**
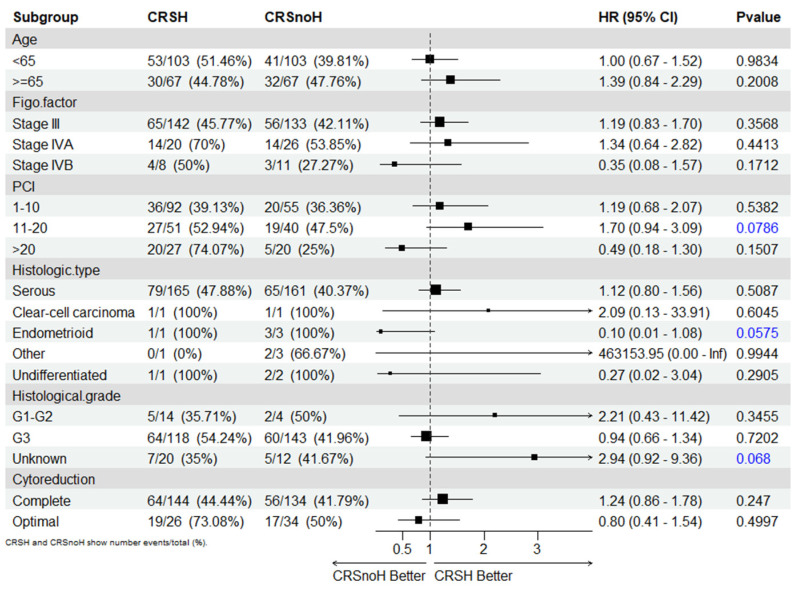
Subgroup analyses for overall survival of the matched series.

**Table 1 cancers-15-04271-t001:** Demographic and baseline disease characteristics of the unmatched cohort.

	CRSnoH n (%)	CRSH n (%)	*p*-Value
	N = 287 (58.7%)	N = 202 (41.3%)	
Age at diagnosis, years,			0.36
mean (SD)	60 (10.21)	61 (10.32)	
Age group			0.29
<65 years	162 (56.4%)	121 (59.9%)	
≥65 years	125 (43.6%)	81 (40.1%)	
Tumor histologic type			0.12
Serous	259 (91.2%)	192 (97.0%)	
Endometroid	9 (3.2%)	1 (0.5%)	
Clear-cell carcinoma	4 (1.4%)	1 (0.5%)	
Undifferentiated	4 (1.4%)	2 (1%)	
Other	8 (2.8%)	2 (1%)	
Histological grade			0
Unknown	20 (7.0%)	47 (23.3%)	
G1–G2	18 (6.3%)	33 (16.3%)	
G3	249 (86.8%)	122 (60.4%)	
FIGO			0
Stage III	153 (53.3%)	164 (81.2%)	
Stage IVA	30 (10.5%)	28 (13.9%)	
Stage IVB	104 (36.2%)	10 (5.0%)	
Comorbidities			0.027
No	166 (58.9%)	97 (48.3%)	
Yes	116 (41.1%)	104 (51.7%)	
Charlson index			<0.001
mean (SD)	1.33 (2.09)	0.49 (1.11)	
Median (P25–P75)	0 (0–2)	0 (0–0)	
CEA			0.33
mean (SD)	4.57 (19.26)	2.94 (6.79)	
Median (P25–P75)	1.20 (0.60–2.40)	2 (1.20–2.73)	
Ca199			0.98
mean (SD)	55.33 (315.14)	54.85(133.85)	
Median (P25–P75)	10 (4–23)	11.50 (6–30.00)	
CA125			<0.001
mean (SD)	2494.09 (5788.41)	613.28 (1423.01)	
Median (P25–P75)	871 (301–2036)	90 (25–595)	
CA153			037
mean (SD)	188.19 (282.18)	125.64 (212.48)	
Median (P25–P75)	76.75 (26.60–152)	80 (38.60–101)	
Categorized Icp			0.35
1–10	170 (59.4%)	110 (54.5%)	
11–20	85 (29.7%)	62 (30.7%)	
>20	31 (10.8%)	30 (14.9%)	
Cytoreduction			0.005
Complete	225 (78.9%)	171 (84.7%)	
Optimal	46 (16.1%)	31 (15.3%)	
Incomplete	14 (4.9%)	0 (0%)	
Postoperative complications			0.9
No	165 (58.9%)	115 (41.1%)	
Yes	117 (58.5%)	83 (41.5%)	
Major complication Dindo-Clavien			0.11
No	151 (52.6%)	108 (53.5%)	
grade I	35 (12.2%)	18 (8.9%)	
grade II	61 (21.3%)	53 (26.2%)	
grade IIIa	10 (3.5%)	11 (5.4%)	
grade IIIb	16 (5.6%)	3 (1.5%)	
grade IVa	5 (1.7%)	6 (3.0%)	
grade IVb	3 (1.0%)	0 (0.0%)	
grade V (death)	2 (0.7%)	2 (1.0%)	
Degree of complication			0.77
No complications	162 (56.6%)	108 (53.7%)	
Low (I–II)	92 (32.2%)	71 (35.3%)	
High (III–IV)	32 (11.2%)	22 (10.9%)	
Reoperation			0.24
No	253 (88.5%)	186 (92.1%)	
Yes	33 (11.5%)	16 (7.9%)	
Postoperative death			1
No	284 (99.3%)	200 (99.0%)	
Yes	2 (0.7%)	2 (1%)	
Postoperative stay (in days)			0.15
mean (sd)	10.31 (10.40)	9.24 (5.47)	
Recurrence at follow-up			0.80
No	83(29.3%)	56 (27.9%)	
Yes	200 (70.7%)	145 (72.1%)	
Type of recurrence			0.12
Peritoneal	53 (26.5%)	36 (24.8%)	
Visceral	25 (12.5%)	16 (11.0%)	
Lymph nodes	16 (8.0%)	22 (15.2%)	
Mixed1 (peritoneal and visceral)	33 (16.5%)	28 (19.3%)	
Mixed 2 (nodal and others)	65 (32.5%)	36 (24.8%)	
NA	8 (4.0%)	7 (4.8%)	
Deceased during follow-up			0.02
No	186 (61.2%)	104 (50.7%)	
Yes	118 (38.8%)	101 (49.3%)	
Time follow-up from diagnosis (months)			<0.001
mean (sd)	34.56 (21.89)	45.34 (32.12)	
Time from surgery to recurrence (months)			<0.001
mean (sd)	14.09 (11.55)	15.60 (12.81)	
Time from surgery to death (months)			0.02
mean (sd)	27.91 (19.03)	34.71 (23.15)	
Time from diagnosis to death (months)			0.16
mean (sd)	32.58 (18.59)	36.58 (22.85)	

**Table 2 cancers-15-04271-t002:** Clinicopathological characteristics of the matched series.

	CRSnoH n (%)	CRSH n (%)	*p*-Value
	N = 170 (50%)	N = 170 (50%)	
Age at diagnosis, years			0.69
mean (SD)	60.32 (10.56)	59.87 (10.21)	
Age group			1
<65 years	103 (60.6%)	103 (60.6%)	
≥65 years	67 (39.4%)	67 (39.4%)	
Tumor histologic type			0.66
Serous	161(94.7%)	165 (96.6%)	
Endometroid	3 (1.8%)	1 (0.6%)	
Clear-cell carcinoma	1 (0.6%)	1 (0.6%)	
Undifferentiated	2 (1.2%)	1 (0.6%)	
Other	3 (1.8%)	1 (0.6%)	
Histological grade			0.007
Unknown	22 (13.6%)	38 (22.4%)	
G1–G2	4 (2.4%)	14 (8.2%)	
G3	143 (84.1%)	118 (69.4%)	
FIGO			0.46
Stage III	133 (78.2%)	142 (83.5%)	
Stage IVA	26 (15.3%)	20 (11.8%)	
Stage IVB	11 (6.5%)	8 (4.7%)	
Comorbidities			0.01
No	106 (63.5%)	84 (49.4%)	
Yes	61 (36.5%)	86 (50.6%)	
Charlson index			<0.001
mean (SD)	1.05 (1.77)	0.47 (1.07)	
Median (P25–P75)	0 (0–2)	0 (0–0)	
CEA			0.25
mean (SD)	5.94 (24.40)	2.93 (7.40)	
Median (P25–P75)	1.28 (0.6–2.67)	2 (1.10–2.70)	
Ca199			0.70
mean (SD)	77.85 (402.48)	61.14 (147.36)	
Median (P25–P75)	9.9 (3.00–23.00)	12 (6.68–39.50)	
CA125			<0.001
mean (SD)	2730.99 (6243.87)	644.93 (1508.49)	
Median (P25–P75)	791 (268–2036)	111 (26–536)	
CA153			0.85
mean (SD)	125.53 (176.2)	143.22 (257.20)	
Median (P25–P75)	59.05 (23.95–134.6)	51 (26.50–101)	
Categorized Icp			0.57
1–10	55 (47.8%)	92 (54.1%)	
11–20	40 (34.8%)	51 (30.0%)	
>20	20 (17.4%)	27 (15.9%)	
Cytoreduction			0.3
Complete	135 (79.4%)	144 (84.7%)	
Optimal	34 (20.0%)	26 (15.3%)	
Incomplete	1 (0.6%)	0 (0%)	
Postoperative complications			0.55
No	90 (48.1%)	97 (51.9%)	
Yes	76 (52.1%)	40 (47.9%)	
Major complication Dindo-Clavien			0.005
No	84 (49.4%)	90 (52.9%)	
grade I	26 (15.3%)	16 (9.4%)	
grade II	34 (20.0%)	43 (25.3%)	
grade IIIa	5 (2.9%)	11 (6.5%)	
grade IIIb	14 (8.2%)	2 (1.1%)	
grade IVa	1 (0.6%)	5 (2.9%)	
grade IVb	2 (1.2%)	0	
grade V (death)	1 (0.6%)	2 (1.2%)	
Degree of complication			0.83
No complications	91 (53.8%)	90 (53.3%)	
minor (I–II)	55 (32.5%)	59 (34.9%)	
major (III–V)	23 (13.6%)	20 (11.8%)	
Reoperation			0.16
No	147 (86.5%)	156 (91.8%)	
Yes	23 (13.5%)	14 (8.2%)	
Postoperative death			1
No	168 (99.4%)	168 (98.9%)	
Yes	1 (0.6%)	2 (1.2%)	
Postoperative stay (in days)			0.048
mean (sd)	11.22 (10.84)	9.34 (5.73)	
Recurrence at follow-up			0.71
No	40(23.5%)	44 (25.9%)	
Yes	130 (76.5%)	126 (74.1%)	
Type of recurrence			0.26
Peritoneal	37 (28.5%)	28 (22.2%)	
Visceral	21 (16.2%)	16 (12.7%)	
Lymph nodes	11 (8.5%)	19 (15.1%)	
Mixed (Peritoneal and Visceral)	19 (14.6%)	23 (18.3%)	
Mixed 2 (Lymph nodes and others)	39 (30.0%)	33 (26.2%)	
NA	3 (2.3%)	7 (5.6%)	
Deceased during follow-up			0.32
No	97 (57.1%)	87 (51.2%)	
Yes	73 (42.9%)	83 (48.8%)	
Time follow-up from diagnosis (months)			0.01
mean (sd)	36.34 (21.74)	43.91 (31.91)	
Time from surgery to recurrence (months)			0.0006
mean (sd)	15.14 (12.51)	15.40 (11.57)	
Time from surgery to death (months)			0.51
mean (sd)	31.27 (20.58)	33.54 (22.50)	
Time from diagnosis to death (months)			0.97
mean (sd)	35.71 (20.25)	35.61 (22.30)	

**Table 3 cancers-15-04271-t003:** Disease-free survival in the time from surgery to relapse.(in months).

Recurrence Free Survival Time (in Months)	Percentil 25 (95% CI)	Median (95% CI)	Percentil 75 (95% CI)
Surgery	8 (7–10)	13 (11–16)	26 (20–42)
Surgery + Hipec	9 (7–11)	16 (13–23)	42 (32–NA)

**Table 4 cancers-15-04271-t004:** Overall Survival Time (in Months).

Overall Survival Time (in Months)	Percentil 25 (95% CI)	Median (95% CI)	Percentil 75 (95% CI)
Surgery	25 (20–35)	51 (41–67)	108 (82–NA)
Surgery + Hipec	27 (24–38)	53 (45–79)	NA (90–NA)

## Data Availability

The data presented in this study are available on request from the corresponding author.

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
