# Peer review of "Neoadjuvant Chemotherapy plus Interval Cytoreductive Surgery with or without Hyperthermic Intraperitoneal Chemotherapy (NIHIPEC) in the Treatment of Advanced Ovarian Cancer: A Multicentric Propensity Score Study"

_cancers, 2023, doi:10.3390/cancers15174271_

Round 1
Reviewer 1 Report
Dear Sirs,
Very good retrospective analisys. Have some week points because lack of randomisation and prospective study. The differencies in study groups are good recalculated and described in the article.
- Table 1. Time from surgery to recurrence (months) mean(sd) 14.09 (11.55) 15.60 (12.81) <0.001 - need to be checked if the "p" value is good counted.
Author Response
Reply to reviewer 1
Dear Sirs,
Very good retrospective analisys. Have some week points because lack of randomisation and prospective study. The differencies in study groups are good recalculated and described in the article.
Answer: Thank you very much for your review of our work. It is indeed a retrospective study, but the statistical technique used (PSM) exquisitely balances the two arms of the study and makes this type of retrospective analysis the closest thing to a randomized RCT.
-Table 1. Time from surgery to recurrence (months) mean(sd) 14.09 (11.55) 15.60 (12.81) <0.001 - need to be checked if the "p" value is good counted.
Answer: we have repeated the analysis and I am afraid that the p value is correct
thank you once again for your work
Antoni Llueca
Reviewer 2 Report
This multicentric retrospective observational study evaluates the possible role of neo-adjuvant chemotherapy plus interval debulking surgery (IDS) with and without hyperthermic intraperitoneal chemotherapy (HIPEC) in the treatment of patients with ovarian cancer unfit for primary cytoreduction.
The paper is well written and the English language is appropriate and understandable.
The clinical arguments presented are aligned with the major recommendations and guidelines of the European and international oncological societies.
Data of this study support the role of HIPEC in terms of feasibility. However it couldn’t provide strong scientific support to consider HIPEC at the time of IDS routinely. Although the statistical analysis was well performed (propensity score matching, by equally balancing the arms), this is a retrospective study with multiple biases that limit its clinical validity.
Most published papers showed significant advantages with better PFS and OS for HIPEC treated patients. Could the Authors explain why no significant OS benefits were observed in their study?
The result could be biased by the inclusion of both platinum sensitive and resistant disease. Could the Authors confirm that the enrolled patients were all platinum responders?
Currently, it has been well established that epithelial ovarian cancer patients include populations with different clinical and biological characteristics. BRCA status plays a significant role on chemo-sensitivity, choice of maintenance treatment, and eventually on PFS and OS. Could the Authors provide data regarding mutational status and maintenance therapies? Were the series of patients who underwent HIPEC and the controls well balanced regarding these significant prognostic factors?
Tables 1 and 2 show demographic and baseline characteristics. CEA and CA19-9 levels in the plasma play a poor role in diagnosis, evaluating response to treatment, and prognosis of serous and endometrioid ovarian cancers. Data on albumin levels could be more significant.
The Authors reported that patients with low-grade tumor were excluded from the study. Table 1 but also Table 2 on clinico-pathological characteristics of the matched series show patients with G1-G2 histological grade. Furthermore, grading was unknown in many patients who underwent HIPEC (more than 20%). More details about these inclusion criteria could be useful.
Table 3 and Abstract/Survival Analysis paragraphs show different p-values (p= 0.031 instead of p= 0.016). The data must be consistent each other.
Author Response
This multicentric retrospective observational study evaluates the possible role of neo-adjuvant chemotherapy plus interval debulking surgery (IDS) with and without hyperthermic intraperitoneal chemotherapy (HIPEC) in the treatment of patients with ovarian cancer unfit for primary cytoreduction.
The paper is well written and the English language is appropriate and understandable.
The clinical arguments presented are aligned with the major recommendations and guidelines of the European and international oncological societies.
First of all, thank you very much for your kind and constructive review. I will try to argue the different considerations that you raise me.
1.- Data of this study support the role of HIPEC in terms of feasibility. However it couldn’t provide strong scientific support to consider HIPEC at the time of IDS routinely. Although the statistical analysis was well performed (propensity score matching, by equally balancing the arms), this is a retrospective study with multiple biases that limit its clinical validity.
Answer:
Of course, retrospective studies have a significant possibility of bias, but for that we have tried to balance the arms of the study 1:1. Keep in mind that this type of statistical study is the closest thing to randomization but with retrospective data. They also offer support for the different RCTs that have already been published and the different meta-analyses that squeeze them to offer us the maximum levels of evidence, including a recent one from our group*. On the other hand, our work shows that despite the denial of the scientific evidence provided by HIPEC in some circles of European gynecology, almost half of our tumors in Spain are operated on by oncological surgeons who do apply HIPEC and gynecologists and oncologists from our beloved Europe/Spain have to make us reflect.
*Llueca M, Ibañez MV, Climent MT, Serra A, Llueca A; MUAPOS and OSRG Working Group. Effectiveness of Hyperthermic Intraperitoneal Chemotherapy Associated with Cytoreductive Surgery in the Treatment of Advanced Ovarian Cancer: Systematic Review and Meta-Analysis. J Pers Med. 2023 Jan 30;13(2):258. doi: 10.3390/jpm13020258. PMID: 36836494; PMCID: PMC9960788.
2.-Most published papers showed significant advantages with better PFS and OS for HIPEC treated patients. Could the Authors explain why no significant OS benefits were observed in their study?
Answer: As you already know, we have tried, statistically speaking, but we have not obtained statistical significance. We believe that it may be due to a lack of n, because it is true that there is a tendency for OS to improve over 6 months in favor of HIPEC.
3.-The result could be biased by the inclusion of both platinum sensitive and resistant disease. Could the Authors confirm that the enrolled patients were all platinum responders?
Answer:
As you know very well, neoadjuvant therapy only supports 3-4 cycles of chemotherapy. If it does not respond after 3 cycles, it is considered non-responder and goes on to consolidation therapy with more chemotherapy or with bevazizumab and therefore they are no longer included in the study.In other words, we avoid this bias.
4.-Currently, it has been well established that epithelial ovarian cancer patients include populations with different clinical and biological characteristics. BRCA status plays a significant role on chemo-sensitivity, choice of maintenance treatment, and eventually on PFS and OS. Could the Authors provide data regarding mutational status and maintenance therapies? Were the series of patients who underwent HIPEC and the controls well balanced regarding these significant prognostic factors?
Answer:
As you have already mentioned, it is a retrospective study of hospital data from different years and it was not routinely performed in all hospitals at the time of the study. The mutational status data was not included in all the records and was not collected so as not to spoil the sample with too many missing. Perhaps it would be important for the future to collect these data on the mutational status and compare the effect of hipec in these patients. We take note for possible studies.
5.-Tables 1 and 2 show demographic and baseline characteristics. CEA and CA19-9 levels in the plasma play a poor role in diagnosis, evaluating response to treatment, and prognosis of serous and endometrioid ovarian cancers. Data on albumin levels could be more significant.
Answer:
You are absolutely right, but as it is a retrospective multicenter study we have not been able to obtain all the variables that we would have liked. The albumin in fact would have given us much more value in the study.
6.-The Authors reported that patients with low-grade tumor were excluded from the study. Table 1 but also Table 2 on clinico-pathological characteristics of the matched series show patients with G1-G2 histological grade. Furthermore, grading was unknown in many patients who underwent HIPEC (more than 20%). More details about these inclusion criteria could be useful.
Answer:
Thanks for your appreciation. As you already know, the determination of stratified nuclear maturation in G1-2-3 has a fairly high concordance index with the definitive biopsy obtained in CRS, for this reason there are a small percentage of patients in whom the initial biopsy indicates a high degree and the final was not concordant. On the other hand, as usual when a tumor has an unknown grade, it is usually because the aggressiveness is such that its necrosis does not allow evaluation of the grade, but they almost always behave like a G3.
7.-Table 3 and Abstract/Survival Analysis paragraphs show different p-values (p= 0.031 instead of p= 0.016). The data must be consistent each other.
Answer:
Thank you for your appreciation, we correct it immediately.
Once again, Thanks for your effort in order to improve our manuscript.
Antoni Llueca
Reviewer 3 Report
This is a nicely designed multicenter retrospective study about the effect of HIPEC after NACT and cytoreductive surgery on overal survival and DFS of advanced EOC patients in Spain. Some suggestions below:
1. As the authors discussed in the manuscript that HIPEC only works for the patients with recent chemotherpy, it raises the question that if different chemotherapy drug used in NACT leads to different results.
2. Please discribe the methods of HIPEC from different centers in the methods section.
3. Contents in tables should be carefully aligned for easier understanding.
Easy to follow, some typos need to be corrected.
Author Response
This is a nicely designed multicenter retrospective study about the effect of HIPEC after NACT and cytoreductive surgery on overal survival and DFS of advanced EOC patients in Spain.
Some suggestions below:
- As the authors discussed in the manuscript that HIPEC only works for the patients with recent chemotherpy, it raises the question that if different chemotherapy drug used in NACT leads to different results.
- Please discribe the methods of HIPEC from different centers in the methods section.
Thank you for your consideration, we have done so.
- Contents in tables should be carefully aligned for easier understanding.
Dear reviewer, thank you very much for your considerations, I will try to answer them.
- As the authors discussed in the manuscript that HIPEC only works for the patients with recent chemotherapy, it raises the question that if different chemotherapy drug used in NACT leads to different results.
Answer:
The two most widely used drugs (cisplatin and paclitaxel) work equally, although pharmacologically cisplatin should work better. I leave you the survival curves (not extracted from the original manuscript in the attached document) to complete my explanation.
- Please discribe the methods of HIPEC from different centers in the methods section.
Answer:
Thank you for your consideration, we have done so.
- Contents in tables should be carefully aligned for easier understanding.
It is true, we will try to ensure that the strict layout to which the journal submits us takes this into account
Once again, Thank you for your work
Antoni Llueca
